29 Jul 2021

# Fractionation of stable carbon isotopes during acetate consumption by methanogenic and sulfidogenic microbial communities in rice paddy soils and lake sediments

Ralf Conrad[1], Pengfei Liu[1,2], Peter Claus[1]

[1]Max Planck Institute for Terrestrial Microbiology, Karl-von-Frisch-Str. 10, 35043 Marburg, Germany

[2]Center for Pan-third Pole Environment, Lanzhou University, Tianshuinanlu 222, Lanzhou, 730000, China

Correspondence to: Ralf Conrad (Conrad@mpi-marburg.mpg.de)

**Running head:** Isotope fractionation by anaerobic acetate consumption

**Abstract.** Acetate is an important intermediate during the degradation of organic matter in anoxic flooded soils and sediments. Acetate is disproportionated to $CH_4$ and $CO_2$ by methanogenic or is oxidized to $CO_2$ by sulfate-reducing microorganisms. These reactions result in carbon isotope fractionation, depending on the microbial species and their particular carbon metabolism. To learn more about the magnitude of the isotopic enrichment factors ($\varepsilon$) involved, acetate conversion to $CH_4$ and $CO_2$ was measured in anoxic paddy soils from Vercelli (Italy) and the International Rice Research Institute (IRRI, the Philippines) and in anoxic lake sediments from the north east (NE) and the south west (SW) basins of Fuchskuhle (Germany). Acetate consumption was measured using samples of paddy soil or lake sediment suspended in water or in phosphate buffer (pH 7.0), both in the absence and presence of sulfate (gypsum), and of methyl fluoride ($CH_3F$), an inhibitor of aceticlastic methanogenesis. Under methanogenic conditions, values of $\varepsilon_{ac}$ for acetate consumption were always in a range of -21‰ to -17‰, but higher in the lake sediment from the SW basin (-11‰). Under sulfidogenic conditions $\varepsilon_{ac}$ values tended to be slightly lower (-26‰ to -19‰) especially when aceticlastic methanogenesis was inhibited. Again, $\varepsilon_{ac}$ in the lake sediment of the SW basin was higher (-18‰ to -14‰). Determination of $\varepsilon_{CH4}$ from the accumulation of $^{13}C$ in $CH_4$ resulted in much lower values (-37‰ to -27‰) than from the depletion of $^{13}C$ in acetate (-21‰ to -17‰), especially when acetate degradation was measured in buffer suspensions. The microbial communities were characterized by sequencing the bacterial 16S rRNA genes as well as the methanogenic *mcrA* and sulfidogenic *dsrB* genes. The microbial communities were quite different between lake sediments and paddy soils, but were similar in the sediments of the two lake basins and in the soils from Vercelli and IRR, and were similar after preincubation without and with addition of sulfate (gypsum). The different microbial compositions could hardly serve for the prediction of the magnitude of enrichment factors.

## 1 Introduction

Organic matter degradation under anaerobic conditions results in the production of $CO_2$, when electron acceptors such sulfate (sulfidogenic conditions) are available, and in the production of $CH_4$ and $CO_2$, when they are absent (methanogenic conditions). Carbon dioxide is not only end product, but can also serve as an intermediate. It can for example be converted by chemolithotrophic microorganisms to $CH_4$ or to acetate. These conversion reactions of $CO_2$ have an isotope effect and result in products that are strongly depleted in $^{13}C$, expressing isotope enrichment factors (ε) being on the order of -70 to -50‰ (Blaser and Conrad, 2016). However, the conversion of acetate to $CO_2$ by sulfidogenic reactions or to $CH_4$ plus $CO_2$ by methanogenic reactions can also have quite substantial enrichment factors, often being on the order of about -20‰ (Goevert and Conrad, 2009; Goevert and Conrad, 2010).

Knowledge of enrichment factors is important for the quantification of the pathways involved in anaerobic degradation of organic matter. For example, the relative contribution of hydrogenotrophic and aceticlastic methanogenesis to total $CH_4$ production can be determined in situ from analytical values of $^{13}C$ in organic matter, $CO_2$ and $CH_4$, if the enrichment factors (ε) are known for the reduction of $CO_2$ to $CH_4$ (hydrogenotrophic methanogenesis) and the cleavage of acetate to $CH_4$ and $CO_2$ (aceticlastic methanogenesis)(Conrad, 2005). While ε of the former reaction can be experimentally determined by incubation in the presence of $CH_3F$ (Chan et al., 2005; Conrad et al., 2007; Holmes et al., 2014; Penning et al., 2006b), that of the latter reaction is not so easy to determine. So far reference values are mainly available from relatively few experiments with pure microbial cultures. This is true for both aceticlastic methanogenic archaea (Gelwicks et al., 1994; Goevert and Conrad, 2009; Krzycki et al., 1987; Penning et al., 2006a; Valentine et al., 2004; Zyakun et al., 1988) and acetate-oxidizing sulfidogenic bacteria (Goevert and Conrad, 2008; Goevert and Conrad, 2010; Londry and DesMarais, 2003).

There are hardly studies of environmental samples, in which ε values of acetate consumption were determined together with the composition of the methanogenic and sulfidogenic microbial communities (Blair and Carter Jr., 1992; Chidthaisong et al., 2002; Conrad et al., 2009; Goevert and Conrad, 2009; Penning et al., 2006a). In order to constrain

the magnitude of such ε values, we decided to investigate the stable carbon isotope fractionation during consumption of acetate in methanogenic and sulfidogenic rice paddy soils and anaerobic lake sediments.

## 2 Materials and Methods

### *2.1 Environmental samples*

The soil samples were from the research stations in Vercelli, Italy and the International Rice research Institute (IRRI) in the Philippines. Sampling and soil characteristics were described before (Liu et al., 2018b). The lake sediments (top 10 cm layer) were from the NE and SW basins of Lake Fuchskuhle (Casper et al., 2003). They were sampled in July 20016 using a gravity core sampler as described before (Kanaparthi et al., 2013).

### *2.2 Paddy soils*

Two different experimental setups were used using soil suspensions in water (unbuffered suspensions) or in 20 mM potassium phosphate buffer (pH 7.0)(buffered suspensions). For the unbuffered suspensions, paddy soil was mixed with autoclaved anoxic $H_2O$ at a ratio of 1:1 and incubated under $N_2$ at 25°C for 4 weeks. Then, 5 ml preincubated soil slurry was incubated at 25°C with 45 ml 5.6 mM sodium acetate in a 150-ml bottle under an atmosphere of $N_2$. The bottles were (i) unamended; (ii) amended with 4.5 ml $CH_3F$; (iii) amended with 200 µl of a gypsum ($CaSO_4.2H_2O$) suspension (corresponding to a concentration of 2.5 M sulfate) giving a final concentration of 10 mM sulfate. The experiment was performed in 4 replicates.

For the buffered suspensions, paddy soil was mixed with autoclaved anoxic $H_2O$ at a ratio of 1:1 and incubated under $N_2$ at 25°C for 4 weeks. In a second incubation, paddy soil was mixed with autoclaved anoxic $H_2O$ at a ratio of 1:1, was amended with 0.07 g $CaSO_4.2H_2O$, and then incubated under $N_2$ at 25°C for 4 weeks. These two preincubated soil slurries were sampled and stored at -20°C for later molecular analysis. The preincubated soil slurries were also used (in 3 replicates) for the following incubation experiments. Three different sets of incubations were prepared. In the first set (resulting in methanogenic conditions), 5 ml soil

slurry preincubated without sulfate was incubated at 25°C with 40 ml 20 mM potassium phosphate buffer (pH 7.0) in a 150-ml bottle under an atmosphere of $N_2$. The bottles were the amended with (i) 5 ml $H_2O$; (ii) 5 ml $H_2O$ + 4.5 ml $CH_3F$; (iii) 5 ml 50 mM sodium acetate; (iv) 5 ml 50 mM sodium acetate + 4.5 ml $CH_3F$. In the second set (resulting in mainly methanogenic conditions), again 5 ml soil slurry preincubated without sulfate was incubated at 25°C with 40 ml 20 mM potassium phosphate buffer (pH 7.0) in a 150-ml bottle under an atmosphere of $N_2$. The amendments were the same as above, but with the addition of 200 µl of a $CaSO_4$ suspension giving a final concentration of 10 mM sulfate. In the third set (resulting in sulfidogenic conditions), 5 ml soil slurry preincubated with sulfate was incubated at 25°C with 40 ml 20 mM potassium phosphate buffer (pH 7.0) in a 150-ml bottle under an atmosphere of $N_2$. The amendments were the same as above, but with the addition of 200 µl of a $CaSO_4$ suspension corresponding to a concentration of 2.5 M (giving a final concentration of 10 mM sulfate).

*2.3 Lake sediments*

For methanogenic conditions, 5 ml lake sediment was incubated in 3 replicates at 10°C with 45 ml 20 mM potassium phosphate buffer (pH 7.0) in a 150-ml bottle under an atmosphere of $N_2$. The bottles were the amended with (i) 5 ml $H_2O$; (ii) 5 ml $H_2O$ + 4.5 ml $CH_3F$; (iii) 5 ml 50 mM sodium acetate; (iv) 5 ml 50 mM sodium acetate + 4.5 ml $CH_3$. Part of the lake sediment was preincubated with sulfate by adding 0.1 g $CaSO_4.2H_2O$ (gypsum) to 50 ml sediment and incubating at 10°C for 4 weeks. For sulfidogenic conditions 5 ml of the preincubated sediment was incubated at 10°C with 40 ml 20 mM potassium phosphate buffer (pH 7.0) in a 150-ml bottle under an atmosphere of $N_2$. The bottles were amended as above, but in addition also with 200 µl of a $CaSO_4$ suspension giving a final concentration of 10 mM sulfate. Samples for later molecular analysis were taken from the original lake sediment and from the lake sediment preincubated with sulfate. The samples were stored at -20°C.

*2.4 Extraction of DNA and amplicon sequencing*

The lake sediments or paddy soils in phosphate buffer were collected by centrifugation (11000 × g, 4°C, 5 min). Genomic DNA were extracted with NucleoSpin Soil Kit (Macherey-Nagel, Düren, Germany) by following the user's manufacture. DNA were checked by gel electrophoresis (1% agarose in TEA buffer, stained with GelRed) and quantified by Qubit 2.0. The amplification of 16S rRNA, *mcrA* and *dsrB* gene were done as described previously (Liu and Conrad, 2017). In brief, First step PCR, for 16S rRNA, h515-Y / h926R primers were used with the following PCR protocol: 94°C for 3 min; 15 cycles with 94°C for 30 s, 52°C for 30 s and 68°C for 60 s; 68°C for 10 min and hold at 8°C. For *mcrA*, hmlas-mod-F / hmcra-rev-R primers were used with the following PCR protocol: 94°C for 4 min; 15-18 cycles with 94°C for 30 s , 60 by 1°C to 55°C for 30 s and 68°C for 60 s; 68°C for 10 min and hold at 8°C. For *dsrB* hDSR1762Fmix / hDSR2010Rmix primers were used with the following PCR protocol: 94°C for 3 min; 25 cycles with 94°C for 30 s, 60°C by 1°C to 50 °C for 30 s and 68°C for 60 s; 68°C for 10 min and hold at 8°C.

In the second step PCR, barcode-head primers were used for the PCR products of 16S rRNA, mcrA and *dsrB* obtained from the first step with the following PCR protocol: 94°C for 3 min; 10-20 cycles with 94°C for 30 s, 52°C for 30 s and 68°C for 60 s; 68°C for 10 min and hold at 8°C.

PCR amplicons were purified by AMP xx for both the first and second step PCR. After quantification, the individual barcoded amplicons of 16S rRNA gene and *dsrB* were mixed in equimolar concentrations, with 16S rRNA gene amplicons added in double amounts. Library was sequenced on an ILLUMINA HISEQ 2000 system using 2×250 cycle combination mode by Max Planck-Genome-Centre (Cologne, Germany). For *mcrA*, individual barcoded amplicons were mixed in equimolar concentrations and library was sequenced on an ILLUMINA MISEQ system using 2 × 300 cycle combination mode by Max Planck-Genome-Centre (Cologne, Germany).

*2.5 Amplicon sequence data processing*

Amplicon Sequence data were analyzed according to pipeline as described previously (Liu and Conrad, 2017). In brief, paired-end reads were first merged by USEARCH and 16S rRNA gene and *dsrB* datasets were separated by primer sequences using CUTADAPT and demultiplexed using QIIME1. Datasets of *mcrA* were demultiplexed using QIIME1directly. All reads were subjected to quality control, *de novo* chimera filtering, singleton filtering and OTU clustering according to the UPARSE pipeline. Species level OTUs for 16S rRNA gene were obtained at 97% sequence identity. Approximate species-level dsrB and *mcrA* OTUs were obtained with the gene-specific OTU threshold 0.90 (Pelikan et al., 2016) and 0.84 (Yang et al., 2014). Taxonomic identities of the OTUs of 16S rRNA gene were assigned with the Ribosomal Database Project Classifier against the SILVA 123 SSU Ref database (Pruesse et al., 2007). Nucleotide sequences of *dsrB* and *mcrA* were initially translated into amino acid sequences using FrameBot (Wang et al., 2013). For *dsrB* gene, amino acid sequences were aligned to the DsrAB reference sequence alignment (Pelikan et al., 2016) using MAFFT (Katoh and Standley, 2013). Subsequently, the taxonomic classification of each *dsrB* OTU was analysed using the Evolutionary Placement Algorithm (EPA) in RAXML (Berger et al., 2011). For *mcrA* gene, amino acid sequences of each OTUs and updated full length *mcrA* amino acid sequences from NCBI were imported into a *mcrA* reference ARB database developed by Angel et al. ( 2012). The taxonomic classification of each *mcrA* OTU was analysed by phylogenetic tree construction using Maximum parsimony implemented in ARB software (Ludwig et al., 2004).

*2.6 Chemical and isotopic analyses*

Chemical and isotopic analyses were performed as described in detail previously (Goevert and Conrad, 2009). Methane was analyzed by gas chromatography (GC) with flame ionization detector. Carbon dioxide was analyzed after conversion to $CH_4$ with a Ni catalyst. Stable isotope analyses of $^{13}C/^{12}C$ in gas samples were performed using GC-combustion isotope ratio mass spectrometry (GC-C-IRMS). Acetate was measured using high-performance liquid chromatography (HPLC) linked via a Finnigan LC IsoLink to an IRMS.

The isotopic values are reported in the delta notation ($\delta^{13}C$) relative to the Vienna Peedee Belemnite standard having a $^{13}C/^{12}C$ ratio ($R_{standard}$) of 0.01118: $\delta^{13}C = 10^3 (R_{sample}/R_{standard} - 1)$. The precision of the GC-C-IRMS was ± 0.2‰, that of the HPLC-IRMS was ± 0.3‰. The carbon of the sodium acetate that was used in the incubation experiments had the following $\delta^{13}C$ values: total acetate, -24.4‰; acetate-methyl, -27.9‰; acetate-carboxyl, -20.9‰.

*2.7 Calculations*

Fractionation factors for reaction A → B are defined after Hayes (Hayes, 1993) as:

$$\alpha_{A/B} = (\delta_A + 1000)/ (\delta_B + 1000) \quad (1)$$

also expressed as $\varepsilon \equiv 1000 (1 - \alpha)$ in permil. The carbon isotope enrichment factor $\varepsilon_{ac}$ associated with acetate consumption was calculated from the temporal change of $\delta^{13}C$ of acetate as described by Mariotti et al. (Mariotti et al., 1981) from the residual reactant

$$\delta_r = \delta_{ri} + \varepsilon [\ln(1- f)] \quad (2)$$

and from the product formed

$$\delta_p = \delta_{ri} - \varepsilon (1 - f) [\ln(1- f)]/f \quad (3)$$

where $\delta_{ri}$ is the isotopic composition of the reactant (acetate) at the beginning, $\delta_r$ is the isotopic composition of the residual acetate and and $\delta_p$ that of the product ($CH_4$), both at the instant when $f$ is determined. $f$ is the fractional yield of the products based on the consumption of acetate ($0 < f < 1$). Linear regression of $\delta^{13}C$ of acetate against $\ln(1 - f)$ yields $\varepsilon_{ac}$ as the slope of best fit lines. Similarly, linear regression of $\delta^{13}C$ of $CH_4$ against $(1 - f) [\ln(1- f)]/f$ yields $\varepsilon_{CH4}$ as the slope of best fit lines. The regressions of $\delta^{13}C$ of acetate were done for data in the range of $f < 0.5$. The linear regressions of $\delta^{13}C$ of $CH_4$ were done either for the entire data range of again only for $f < 0.5$. The linear regressions were done individually for each experimental replicate (n = 3-4) and were only accepted if $r^2 > 0.8$ for paddy soils or $r^2 > 0.7$ for lake sediments. The ε values resulting from the replicate experiments were then averaged (± SE).

For mass balance calculations, total inorganic carbon (TIC) was calculated as the sum of gaseous $CO_2$, dissolved $CO_2$ and bicarbonate using the measured data of gaseous $CO_2$, the pH and the relevant solubility and equilibrium constants (Stumm and Morgan, 1996).

## 3 Results

### *3.1 Incubation of unbuffered suspensions of rice field soils*

Incubation of unbuffered suspensions (soil + $H_2O$) of rice field soil from the International Rice Research Institute (IRRI) of the Philippines with acetate under anoxic conditions resulted in the depletion of acetate and the release of $CH_4$ and $CO_2$ (Fig. 1A, C). In the presence of $CH_3F$, an inhibitor of aceticlastic methanogenesis (Janssen and Frenzel, 1997), acetate was no longer consumed, and production of $CH_4$ and $CO_2$ was inhibited. However, addition of sulfate had only little effect on acetate consumption and the production of $CH_4$ and $CO_2$ (Fig. 1A, C). Both in the presence and the absence of sulfate, $\delta^{13}C$ of the residual acetate and the produced $CH_4$ increased, whereas $\delta^{13}C$ in $CO_2$ stayed relatively stable (Fig. 1B, D). In the presence of $CH_3F$, the $\delta^{13}C$ in $CH_4$ was much more negative than in the absence. The results were similar for soil from the Rice Research Station in Vercelli (Italy) (Fig.S1).

Mariotti plots of the $^{13}C$ of acetate as function of the fractions (*f*) of acetate consumed resulted in similar curves for all four replicates of the incubations of IRRI soil without (Fig. 2A) and with (Fig. 2B) sulfate amendment. The lines were straight for $f < 0.5$ (<50% of acetate consumed). The same was the case for Mariotti plots of $^{13}CH_4$, the product of acetate consumption (Fig. 2C, D). The enrichment factors ε, which were calculated from the Mariotti plots were in a range of -22‰ to -19‰, irrespectively whether they were determined in the presence or the absence of sulfate and whether they were determined from acetate depletion or from $CH_4$ formation (Fig. 3; Table S1). Similar Mariotti plots were obtained for Vercelli soil (Fig. S2), which resulted in ε values ranging between -20‰ to -17‰, except the ε determined for $CH_4$ production in the presence of sulfate, which was only -14 ± 1.4‰ (Fig. 3).

Mass balance calculations showed that on a molar basis the accumulated $CH_4$ amounted to about 90% of the consumed acetate (the methyl group) in the absence and to about 71% in the presence of sulfate in the IRRI soil and to 97% and 76%, respectively, in the Vercelli soil.

*3.2 Incubation of buffered suspensions of rice field soils*

The experiments with rice field soils were repeated using soil slurries suspended in phosphate buffer. This was done to run the experiment at a constant pH 7.0. In Vercelli soil, acetate was consumed and $CH_4$ and $CO_2$ were produced (Fig. 4A, C, E). The $\delta^{13}C$ of the residual acetate and the produced $CH_4$ increased as acetate consumption proceeded (Fig. 4B, D). The $\delta^{13}C$ of the produced $CO_2$ first decreased and later increased (Fig. 4F). This happened also, when the soil suspensions were incubated in the presence of sulfate after preincubation with sulfate (Fig. 4F), but $CH_4$ production was lower and $CO_2$ production was higher in the presence than in the absence of sulfate (Fig. 4C, E). Production of $CH_4$ ceased in the presence of sulfate after about 10 d (Fig. 4C). Addition of $CH_3F$ completely inhibited $CH_4$ production both in the presence and absence of sulfate (Fig. 4C). It also inhibited $CO_2$ production but only in the absence of sulfate (Fig. 4E). In the presence of sulfate, $CH_3F$ only delayed but did not inhibit acetate consumption and $CO_2$ production (Fig. 4A, E), and also did not prevent the increase of $\delta^{13}C$ in the residual acetate (Fig. 4B).

Mass balance calculations showed that while acetate (the methyl group) was almost completely degraded to $CH_4$ in the absence of sulfate, it accounted, after a delayed inhibition, for only about a third in the presence of sulfate (Fig. 5). When the soil suspensions were incubated in the presence of sulfate but without preincubation, $CH_4$ production was only slightly less than in the incubations without sulfate (Fig. 5). The complete set of these experiments is shown in Fig. S3. When $CH_4$ production was inhibited, $CO_2$ production apparently was a substitute, since the consumed acetate was always rather well balanced by the production of both $CH_4$ and TIC together (Fig. S4). The same experimental setup was used for IRRI soil. The results were similar and are shown in the supplementary (Fig. S5, S6, S7). In IRRI soil suspensions, the mass balance between the production of $CH_4$ + TIC and the acetate consumed was improved when the production was corrected with the background production in a control without addition of acetate (Fig. S8).

Mariotti plots of acetate consumption and $CH_4$ production in both Vercelli soil (Fig. S9) and IRRI soil (Fig. S10) could be created for all the different incubation conditions, in which acetate was consumed, i.e. in the absence of sulfate (control), in the presence of sulfate, and in

the presence of sulfate after preincubation with sulfate. Enrichment factors (ε) were calculated for fractions of acetate consumption with *f* < 0.5 (Fig. 3; Table S1). The ε values for acetate consumption were similar for the experiments without and with sulfate and ranged between -21‰ and -17‰. However, the ε values for $CH_4$ production were systematically lower, ranging between -37‰ and -23‰ (Fig. 3; Table S1). Since acetate consumption in the presence of sulfate was also possible when $CH_4$ production was inhibited by $CH_3F$, Mariotti plots could also be created for these conditions (Fig. S11). The resulting ε values were similar than those in the absence of sulfate and ranged for Vercelli soil between -24‰ and -22‰ (Fig. 3; Table S1). Only in the IRRI soil ε values were higher (-10‰), but only when the soil had been preincubated with sulfate (Fig. 3; Table S1).

*3.3 Incubation of buffered suspensions of lake sediments*

Experiments with lake sediments were done analogous to those with rice filed soils. Slurries of sediment from the NE and SW basins of Lake Fuchskuhle were suspended in phosphate buffer pH 7.0 in the absence and the presence of sulfate (after preincubation with sulfate) and without and with addition of $CH_3F$. In the sediment from the NE basin acetate was consumed after a lag phase, first (after about 40 d) in the incubations with sulfate, then (after about 60 d) also in the incubations without sulfate (Fig. 6A). Addition of $CH_3F$ only partially inhibited the acetate consumption in the absence of sulfate, and did not at all inhibit the acetate consumption in the presence of sulfate (Fig. 6A). However, $CH_3F$ almost completely inhibited the production of $CH_4$, and also inhibited almost completely the increase of the $\delta^{13}C$ in the residual acetate when sulfate was absent (Fig. 6B). Presence of sulfate also strongly inhibited $CH_4$ production (Fig. 6C). The small amounts of $CH_4$ produced showed a rather constant $\delta^{13}C$ of about -40‰ in the absence and of -90 to -80‰ in the presence of $CH_3F$ (Fig. 6D). Without sulfate, by contrast, the $\delta^{13}C$ in $CH_4$ was first about -70‰ and then with acetate consumption progressively increased to about -40‰ in the absence and decreased to about -90‰ in the presence of $CH_3F$ (Fig. 6D). Mass balance calculations showed that $CH_4$ production in the presence of sulfate accounted on a molar basis only for about 5% of the acetate consumed, while in the absence of sulfate $CH_4$ production accounted for about 45%

(Fig. 5). In the sediment from the SW basin, the contribution of $CH_4$ production to acetate consumption was even lower (about 30%)(Fig. 5). These low values are noteworthy in comparison to those found in the rice field soils (Fig. 5). In the presence of sulfate, acetate was almost exclusively converted to $CO_2$, which strongly increased during the time of acetate consumption exhibiting a relatively good mass balance (Fig. S4, S8). This was also the case in the incubations without sulfate, indicating that a rather large fraction of the acetate was converted to $CO_2$ rather than $CH_4$ (Fig. S4, S8). The $\delta^{13}C$ of the produced $CO_2$ strongly decreased from about -30‰ to about -55‰ during the period of acetate consumption (Fig. 6F) and then slowly increased back to -30‰, when about 50% of the acetate had been consumed (Fig. 6A). The experimental results were similar in the incubations with sediment from the SW basin, which are shown in the supplement (Fig. S12, S13).

Mariotti plots of acetate consumption could be generated for all incubation conditions both in the NE and in the SW basin of Lake Fuchskuhle (Fig. S14, S15). These plots allowed the calculation of ε values, which were generally higher (-20‰ to -19‰) in the NE (-20‰ to -19‰) than the SW basin (-14‰ to -11‰). For $CH_4$ production, useful Mariotti plots could only be generated for incubations without sulfate resulting in ε values, which were higher (-28‰ to -27‰) than those calculated from acetate consumption (Fig. 3; Table S1). Mariotti plots of acetate consumption could be generated for incubations with sulfate, in which the very low $CH_4$ production was further inhibited by $CH_3F$ (Fig. S16). The ε values of these incubations (NE and the SW basin, respectively) were lower (-26‰, and -24‰) than those without $CH_3F$ (-20‰ and -14‰) (Fig. 3; Table S1).

### *3.4 Microbial community composition*

The composition of the microbial communities was determined at the beginning of incubation, after preincubation without and with sulfate in the rice field soil and lake sediment suspensions by targeting three different genes, i.e., *mcrA* (methyl CoM reductase), *dsrB* (dissimilatory sulfate reductase), and the bacterial 16S rRNA gene. The compositions of microorganisms represented by all three genes were quite different between the rice field soils

and the lake sediments, while the differences within the individual samples of either soils or sediments were smaller (Fig. 7).

In the sediments of both basins of Lake Fuchskuhle, the methanogenic archaea (represented by *mcrA*) were dominated by *Methanomicrobiales* and *Methanosaetaceae*, while *Methanomassiliicoccales* and *Methanosarcinaceae* contributed less (Fig. 7A). In the rice field soils, the methanogenic taxa were more diverse comprising 6 different orders or families, with putatively aceticlastic *Methanosarcinaceae* being relatively more abundant than *Methanosaetaceae*. In addition, *Methanomicrobiales* contributed only little compared to *Methanobacteriales* and especially *Methanocellales*. In general, there was only a marginal difference in composition between the incubations in the absence and the presence of sulfate.

The composition of putative sulfate reducers (represented by *dsrB*) was also only little different between the incubations with and without sulfate addition (Fig. 7B). However, the composition between rice field soils and lake sediments was completely different. While rice field soils were dominated by members of the uncultured-family-level lineages 9 and 5, the lake sediments were dominated by the *Desulfobacca acetoxidans* lineage. Compared to IRRI soil the relative abundance of *Syntrophobacteraceae* was larger in Vercelli soil, where it increased upon treatment with sulfate. In the lake sediments, the relative abundances of *Syntrophobacteraceae* and members of environmental superclusters were similar. In the lake sediments there was only some minor quantitative difference between the two basins, while the difference between Vercelli and IRRI soil was more pronounced. For example, compared to IRRI soil the relative abundance of *Syntrophobacteraceae* was larger in Vercelli soil, where it increased upon preincubation with sulfate.

The composition of Bacteria in general (represented by the 16S rRNA gene) was again most different between rice field soils and lake sediments, while differences between Vercelli and IRRI soils and also between sediments from the NE and SW lake basins were much less, and differences between preincubations without and with sulfate were marginal (Fig. 7C). While in rice field soils Clostridia were the most abundant group followed by Deltaproteobacteria, it was the other way round in the lake sediments. Rice field soils

contained *Bacilli*, while Lake sediments contained *Spirochaetes*, which were respectively negligible.

## 4 Discussion

### *4.1 Methanogenic conditions*

We measured $\varepsilon_{ac}$ values in anaerobic environmental samples, which consumed acetate almost exclusively by methanogenesis. Predominance of methanogenesis occurred in the absence of sulfate, in some incubations of the rice field soils even in the presence of sulfate provided there was no prior incubation in the presence of sulfate. In the rice field soils, $CH_4$ carbon accounted for more than 90% of the consumed acetate carbon, and $CH_3F$ completely inhibited acetate consumption, the increase of $\delta^{13}C$ in the residual acetate, and also inhibited most of $CH_4$ production. In conclusion, acetate was exclusively consumed by aceticlastic methanogenesis and only little $CH_4$ was produced from other sources than acetate, presumably from background organic carbon via hydrogenotrophic methanogenesis as indicated by the negative $\delta^{13}C$ of the produced $CH_4$. The increase of $\delta^{13}C$ in the residual acetate was expected due to preferred utilization of isotopically light acetate carbon. Such patterns of $CH_4$ production and change in $^{13}C$ isotopic signatures have been observed by us before in rice field soils and lake sediments (Conrad et al., 2010; Conrad et al., 2009; Fu et al., 2018; Ji et al., 2018). They are in agreement with the presence of a diverse methanogenic archaeal community consisting of putatively hydrogenotrophic and aceticlastic methanogenic archaea, which have been found in both Vercelli and IRRI soils (Liu et al., 2019; Liu et al., 2018b). The aceticlastic methanogens consisted of species of the genera *Methanosarcina* and *Methanosaeta* (or *Methanothrix* (Oren, 2014)), which differ in the mechanism of acetate activation and the affinity towards acetate (Jetten et al., 1990).

Both genera of methanogens were also present in the sediments of Lake Fuchskuhle, which exhibited a similar pattern of acetate consumption and $CH_4$ production as the anaerobic rice field soils, thus confirming and extending earlier studies (Chan et al., 2002; Conrad et al., 2010). Notably, $CH_3F$ addition again almost completely inhibited $CH_4$ production from acetate and was accompanied by highly negative $\delta^{13}C$ in the small amounts of residual $CH_4$,

which was presumably produced by hydrogenotrophic methanogenesis. Addition of $CH_3F$ did not completely inhibit acetate consumption, indicating consumption by oxidation rather than aceticlastic methanogenesis. However, $CH_3F$ almost completely inhibited the increase of $\delta^{13}C$ in the residual acetate, indicating only a negligible isotope effect. Also, $CH_4$ production accounted only for less than 50% of the consumed acetate, as production of $CH_4$ was replaced by $CO_2$. Hence, part of the acetate was apparently consumed by oxidative processes, even without addition of sulfate. We assume that the oxidation consumption process was driven by humic acids (Lovley et al., 1996). Notably, imbalance in the stoichiometry between $CH_4$ and consumed acetate is reflected by the fact that the SW basin has a higher humic acid content than the NE basin (Casper et al., 2003).

Values of $\varepsilon_{ac}$ measured in cultures of methanogenic archaea differ depending on the genus and the corresponding mechanism of acetate activation. Thus, methanogenic archaea of the genus *Methanosarcina*, which activate acetate with acetate kinase and phosphotransacetylase have a relatively negative $\varepsilon_{ac}$ with values ranging between -35‰ and -21‰ (Gelwicks et al., 1994; Goevert and Conrad, 2009; Krzycki et al., 1987; Zyakun et al., 1988). By contrast, those of the genus *Methanosaeta*, which activate acetate with acetyl-CoA synthetase, have less negative $\varepsilon_{ac}$ with values ranging between -14‰ and -10‰ (Penning et al., 2006a; Valentine et al., 2004). The $\varepsilon_{ac}$ values in methanogenic rice field soils were all in a range of -21‰ to -17‰, which is at the less negative end or even a bit less negative than the values reported for pure cultures of *Methanosarcina* but is more negative than the values reported for *Methanosaeta*. Therefore, it is reasonable to conclude that in the methanogenic rice field soils, acetate was consumed mainly by *Methanosarcina* species and only to a minor extent by *Methanosaeta* species. This conclusion is in agreement with the composition of the soil methanogenic archaeal communities, which consisted of both genera. A similar conclusion has been reached in methanogenic rice field soil (Goevert and Conrad, 2009). A similar enrichment factor for acetate consumption has also been measured in the anoxic sediment of Lake Wintergreen (Gelwicks et al., 1994) and again in the present study of the NE basin of Lake Fuchskuhle. However, the sediment of the SW basin of Lake Fuchskuhle exhibited a less negative $\varepsilon_{ac}$ of about -11‰, which would be consistent with the activity of aceticlastic

*Methanosaeta* species. Indeed, *mcrA* genes of *Methanoseata* species were much more abundant in the sediments of Lake Fuchskuhle than *mcrA* genes of *Methanosarcina* species. Also in Lake Dagow sediments (located in the same region of Germany), methanogenic archaea were dominated by *Methanosaeta* species and exhibited a relatively high $\varepsilon_{ac}$ of about -13‰ (Penning et al., 2006a). However, the sediment of the NE basin of Lake Fuchskuhle, which was also dominated by *Methanosaeta* species exhibited more negative $\varepsilon_{ac}$ values of about -19‰.

Methanogenic consumption results in disproportionation of the acetate molecule with oxidation of the carboxyl group to $CO_2$ and reduction of the methyl group to $CH_4$. In context of the isotope fractionation during the conversion of acetate to $CH_4$ it is the isotopic enrichment factor of the methyl group, $\varepsilon_{ac\text{-}methyl}$, which matters. Studies of fractionation of the acetate-methyl in pure culture studies of aceticlastic methanogenic archaea have shown that $\varepsilon_{ac\text{-}methyl}$ was always a few permil less negative than $\varepsilon_{ac}$. This difference was due to a larger isotope effect for the conversion of acetate-carboxyl than acetate-methyl (Gelwicks et al., 1994; Goevert and Conrad, 2009; Penning et al., 2006a; Valentine et al., 2004). Alternatively to $\varepsilon_{ac\text{-}methyl}$ the enrichment factor for the conversion of acetate-methyl to $CH_4$ can also be measured from the isotopic composition in $CH_4$, i.e., $\varepsilon_{CH4}$. Most of the studies of pure methanogenic cultures resulted in $\varepsilon_{CH4}$ being similar to $\varepsilon_{ac\text{-}methyl}$ (Gelwicks et al., 1994; Goevert and Conrad, 2009; Penning et al., 2006a), but occasionally $\varepsilon_{CH4}$ was a few permil more negative than $\varepsilon_{ac\text{-}methyl}$, both in pure culture (Valentine et al., 2004) and in environmental samples (Goevert and Conrad, 2009). Similarly, values of $\varepsilon_{CH4}$ in the unbuffered suspensions of rice field soils were only slightly more negative than values of $\varepsilon_{ac}$. However, in the buffered suspensions of both rice field soils and lake sediments, values of $\varepsilon_{CH4}$ were much more negative than those of $\varepsilon_{ac}$, the difference amounting to 9-17‰. These results indicate that the isotope effect for the conversion of the acetate-methyl to $CH_4$ was much stronger than that for the conversion of acetate-carboxyl to $CO_2$, which is completely opposite to the results obtained in cultures of methanogenic archaea. This discrepancy in the results is presently without conclusive explanation. The possibility of effects by bicarbonate or $CO_2$ concentrations, being different in the pure microbial cultures, the unbuffered and buffered soil

suspensions, or of phosphate effects should be considered. Effects of $CO_2$ concentrations and buffer systems on fractionation factors have for example been observed in cultures of chemolithoautotrophic *Thermoanaerobacter kivui* (Blaser et al., 2015).

*4.2 Sulfidogenic conditions*

We also measured $\varepsilon_{ac}$ values in anaerobic environmental samples, which consumed acetate by sulfate reduction. These conditions were achieved (only in the buffered suspensions) by preincubation with gypsum and measurement of acetate consumption in the presence of sulfate (gypsum). Preincubation was required because of delayed sulfate reduction (Liu et al., 2018a). The relative abundance of both *dsrB* genes and genes of bacterial 16S rRNA were only marginally different between samples preincubated under methanogenic and sulfidogenic conditions, similarly as observed before (Wörner et al., 2016). It is probably the induction of the sulfate reduction activity, which delayed sulphidogenic conditions (Liu et al., 2018a). Sulfidogenic conditions were verified by showing that methanogenesis was almost completely inhibited while acetate consumption operated, also with respect to increase of $\delta^{13}C$ in the residual acetate, and that $CH_3F$, which is rather specific for acetoclastic methanogenesis, had only a marginal effect on these sulfidogenic activities. Finally, sulfidogenic conditions were verified by the stoichiometry of acetate conversion, which showed only very little $CH_4$ production.

Values of $\varepsilon_{ac}$ measured in cultures of sulfate-reducing bacteria differ depending on the genus and the corresponding mechanism of acetate dissimilation. Experiments with cultures of sulfate reducers showed that *Desulfobacca acetoxidans*, which dissimilates acetate via the acetyl-CoA pathway, exhibits of about -19‰ being similar to the $\varepsilon_{ac}$ values of aceticlastic *Methanosarcina* species. By contrast, cultures of *Desulfobacter* species, which dissimilate acetate vie the tricarbonic acid cycle, exhibited $\varepsilon_{ac}$ values of about +2‰ (Goevert and Conrad, 2008). The $\varepsilon_{ac}$ values measured in sulfidogenic anoxic paddy soils were in a range of -24‰ to -22‰, but were only -24‰ to -10‰, when measured in the presence of $CH_3F$, which guarantees that all aceticlastic methanogenic activities were inhibited. This range of $\varepsilon_{ac}$ values compares rather well with the value of -19‰ measured in *Desulfobacca acetoxidans*, which

however, was of only low relative abundance in the rice field soil incubations. However, it is well possible that uncultured-family-level lineages, which were the major sulfate reducers, dissimilated by similar pathway than *Desulfobacca acetoxidans* and thus, exhibited similar $\varepsilon_{ac}$ values. Also *Syntrophobacteraceae,* which have been found to act as major acetate-utilizing sulfate reducers in Vercelli soil (Liu et al., 2018a) increased in relative abundance after preincubation with sulfate. The lake sediments, by contrast, exhibited a high relative abundance of *Desulfobacca acetoxidans*, and $\varepsilon_{ac}$ values (including those with $CH_3F$) were in a range of -26‰ to -14‰. Unfortunately, there is, to our knowledge, only a paucity of $\varepsilon_{ac}$ values measured in cultures of sulfate reducers (Goevert and Conrad, 2008). Therefore, it is not possible to have a better resolution of the role of different taxa and metabolic types of sulfate-reducing bacteria on the fractionation of acetate carbon.

## 5 Conclusions

In order to learn about the factors that affect the magnitude of $^{13}C$ isotope fractionation during anaerobic acetate consumption, we studied acetate consumption under methanogenic and sulfidogenic conditions in four different environmental samples, two rice field soils and two lake sediments, by quantifying the conversion of acetate to $CH_4$ and $CO_2$ and by measuring the $\delta^{13}C$ in these compounds, and also determined the composition of the microbial communities. Despite a relatively wide variety of environmental conditions and microbial community compositions, the range of fractionation factors (isotopic enrichment factors $\varepsilon_{ac}$ for the fractionation of total acetate) was quite moderate. The observed $\varepsilon_{ac}$ values were basically within the range that is known from studies of pure cultures of sulfate-reducing bacteria and methanogenic archaea, with a predominance of $\varepsilon_{ac}$ values around -20‰, which is consistent with acetate fractionation in both aceticlastic *Methanosarcina* species and acetate-dissimilating sulfate reducers using the acetyl-CoA pathway. In few cases $\varepsilon_{ac}$ values were close to -10‰, being consistent with a predominance of aceticlastic *Methanosaeta* species. However, there is a paucity of data from cultures of acetate-dissimilating sulfate reducing bacteria, for example from *Syntrophobacteraceae*, which presently limits the potential for predicting fractionation of acetate carbon by knowing the microbial community composition.

Another point of concern is the use of buffered growth media, which may affect isotope fractionation, such as indicated by the observation that $\varepsilon_{CH4}$ values were much more negative than $\varepsilon_{ac}$ values when using suspensions in phosphate buffer rather than in water.

**Acknowledgements**

We thank Dres Peter Casper and Dheeraj Kanaparthi for providing sediment samples from Lake Fuchskuhle. We thank the Fonds der Chemischen Industrie Deutschland for financial support.

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

**Figure legends**

Fig. 1: Acetate conversion to $CH_4$ and $CO_2$ in unbuffered suspensions of paddy soil from the IRRI (the Philippines) without additions (water control), with gypsum, and with $CH_3F$. The panels show the temporal change of (A) concentrations of acetate; (B) $\delta^{13}C$ of acetate; (C) partial pressures of $CH_4$ and $CO_2$ (1 ppmv = $10^{-6}$ bar); (D) $\delta^{13}C$ of $CH_4$ and $CO_2$. Means ± SE, n = 2.

Fig. 2: Mariotti plots of (A, B) acetate consumption and (C, D) $CH_4$ production in (A, C) the absence (control) and (B, D) the presence of gypsum (+ sulfate) in 4 replicates of unbuffered suspensions of paddy soil from the IRRI.

Fig. 3: Isotopic enrichment factors ($\varepsilon_{ac}$ or $\varepsilon_{CH4}$, given as negative values) of unbuffered soil suspensions (Vercelli-soil, IRRI-soil) and buffered suspensions of paddy soil (Vercelli, IRRI) or sediments of Lake Fuchskuhle (NE and SW basin). The values of $\varepsilon_{ac}$ (acetate) and $\varepsilon_{CH4}$ ($CH_4$) were measured without addition of sulfate (methanogenic conditions), with addition of sulfate during preincubation and the experiment (sulfidogenic conditions) and with sulfate but the preincubation without sulfate (mostly methanogenic conditions). Mean ± SE, n = 3-4.

Fig. 4: Acetate conversion to $CH_4$ and $CO_2$ in phosphate-buffered (pH 7.0) suspensions of paddy soil from Vercelli (Italy) without additions (control); with $CH_3F$; with gypsum (preincubation and experiment); with gypsum (preincubation and experiment) + $CH_3F$. The panels show the temporal change of (A) concentrations of acetate; (B) $\delta^{13}C$ of acetate; (C) partial pressures of $CH_4$ (1 ppmv = $10^{-6}$ bar); (D) $\delta^{13}C$ of $CH_4$; (E) partial pressures of $CO_2$ (1 ppmv = $10^{-6}$ bar); (D) $\delta^{13}C$ of $CO_2$. Means ± SE, n = 3.

Fig. 5: Balance of $CH_4$ produced against acetate consumed in phosphate-buffered suspensions of paddy soil from Vercelli and IRRI, and of sediments from the NE and SW basin of Lake Fuchskuhle. The figures show individual replicates (n = 3) of the unamended control (methanogenic conditions); of the experiment plus gypsum ($CaSO_4$-1); of preincubation and experiment plus gypsum ($CaSO_4$-2). The diagonal line indicates stoichiometric conversion (disproportionation) of acetate to $CH_4$ + $CO_2$.

Fig. 6: Acetate conversion to $CH_4$ and $CO_2$ in phosphate-buffered (pH 7.0) suspensions of sediment from the NE basin of Lake Fuchskuhle without additions (control); with $CH_3F$; with gypsum (preincubation and experiment); with gypsum (preincubation and experiment) + $CH_3F$. The panels show the temporal change of (A) concentrations of acetate; (B) $\delta^{13}C$ of acetate; (C) partial pressures of $CH_4$ (1 ppmv = $10^{-6}$ bar); (D) $\delta^{13}C$ of $CH_{4;}$ (E) partial pressures of $CO_2$ (1 ppmv = $10^{-6}$ bar); (D) $\delta^{13}C$ of $CO_2$. Means ± SE, n = 3.

Fig. 7: Relative abundance of (A) *mcrA* (methanogens), (B) *dsrB* (sulfate reducers), (C) bacterial 16S rRNA genes; The DNA was extracted after preincubation of phosphate-buffered suspensions of paddy soils (Vercelli, IRRI) and sediments of Lake Fuchskuhle (NE, SW basin) without additions (methanogenic conditions) or preincubated and incubated in the presence of sulfate (sulfidogenic conditions).

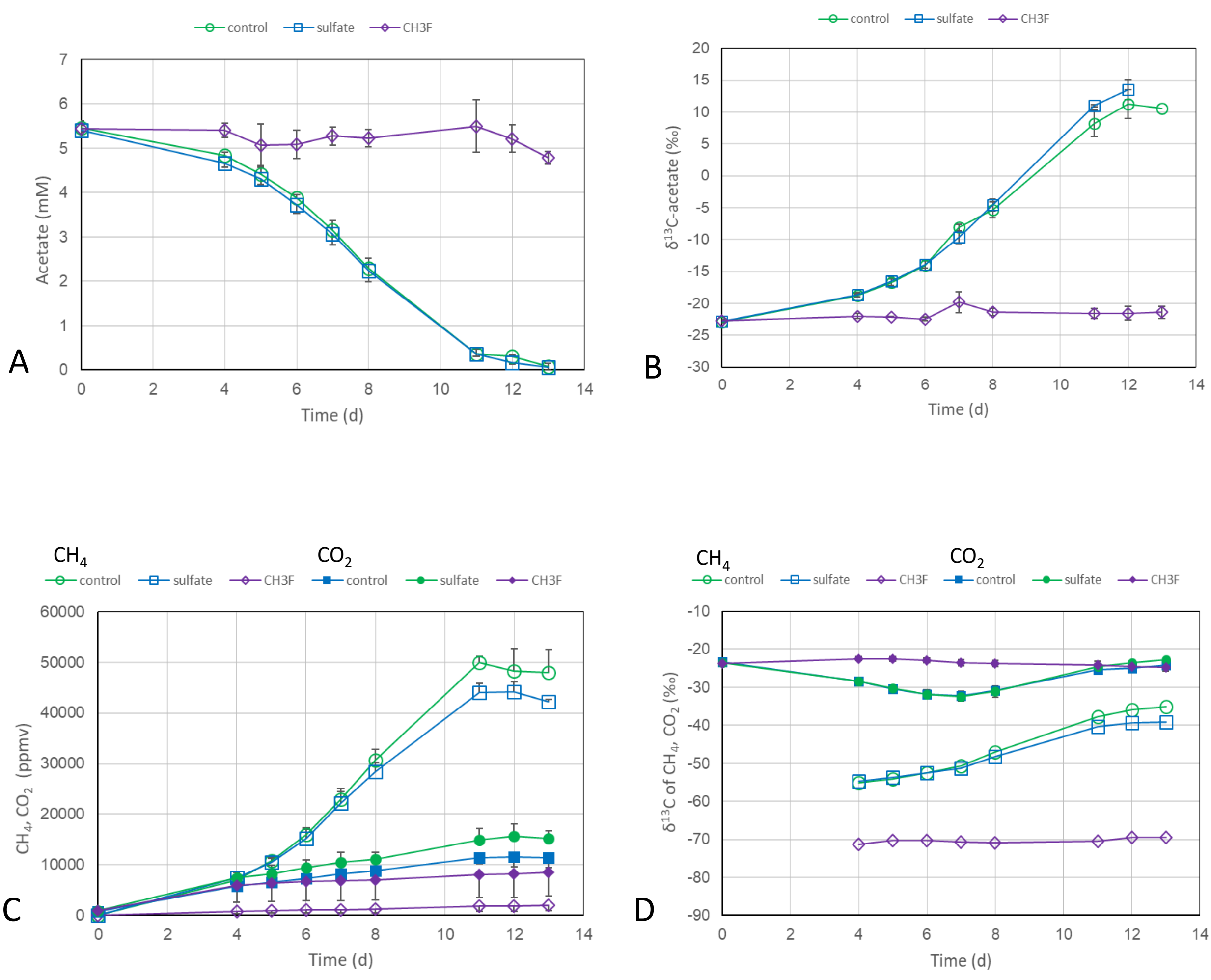

Fig. 1

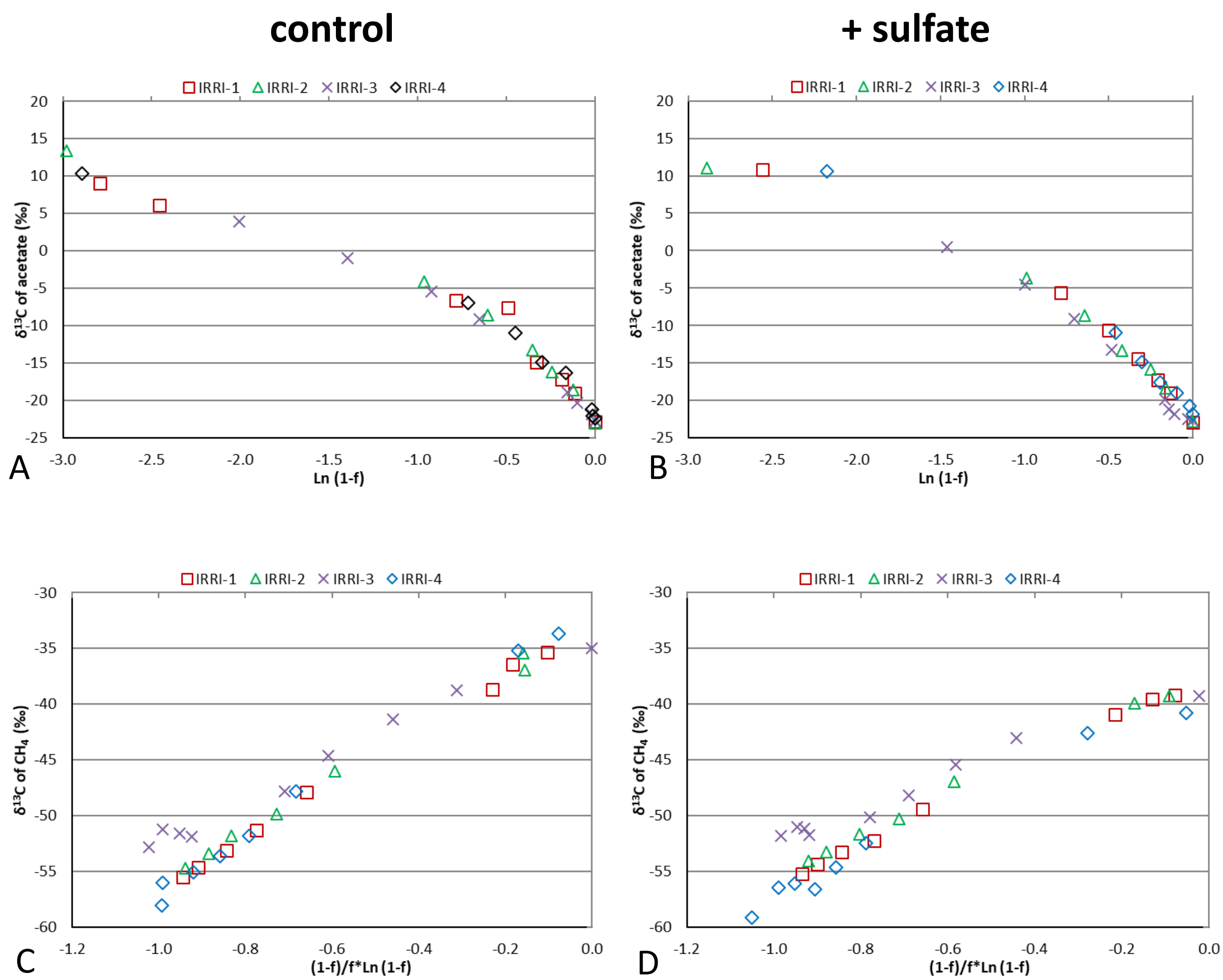

Fig. 2

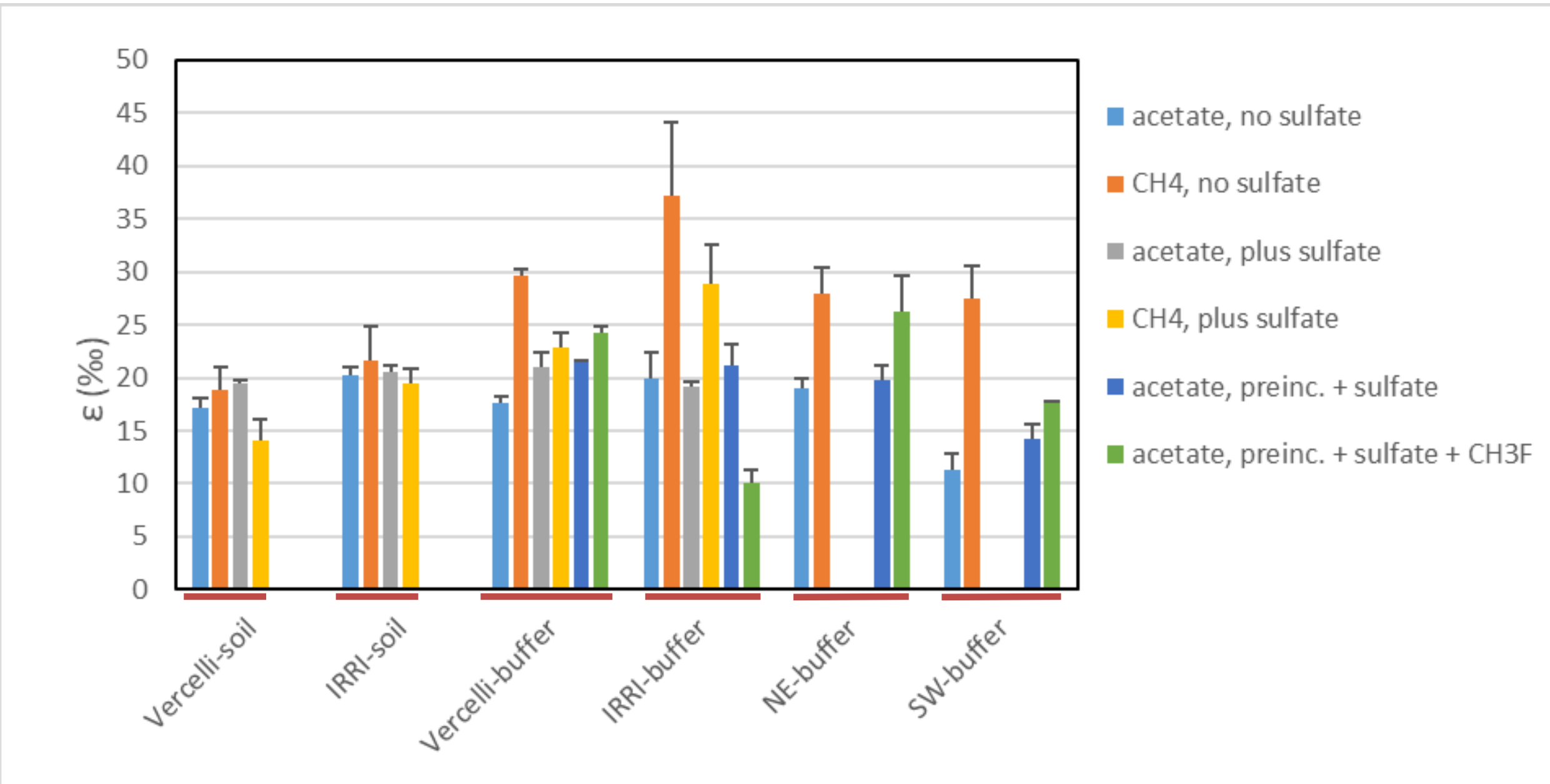

Fig. 3

Open Access EGU

Vercelli, 2nd experiment

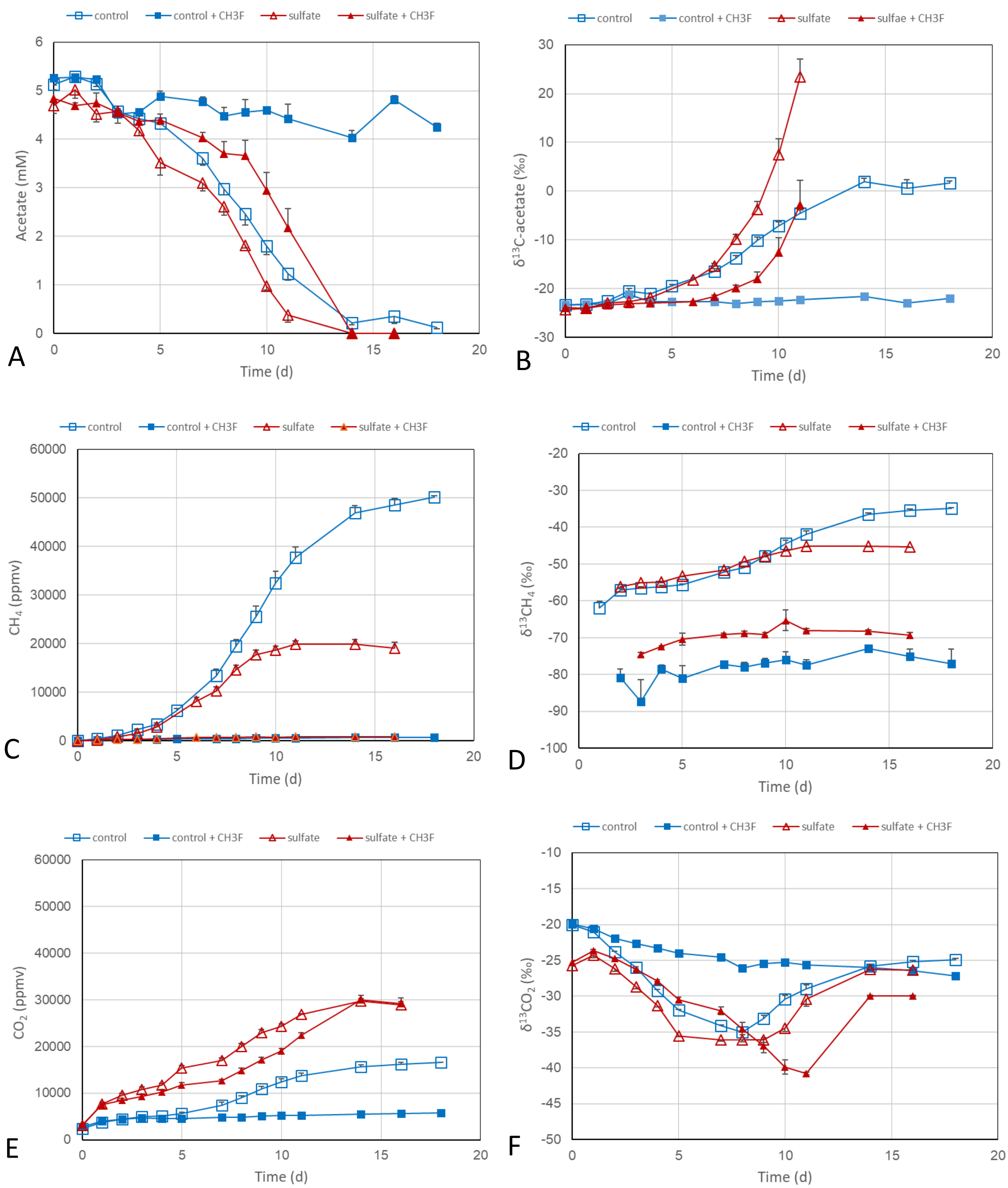

Fig. 4

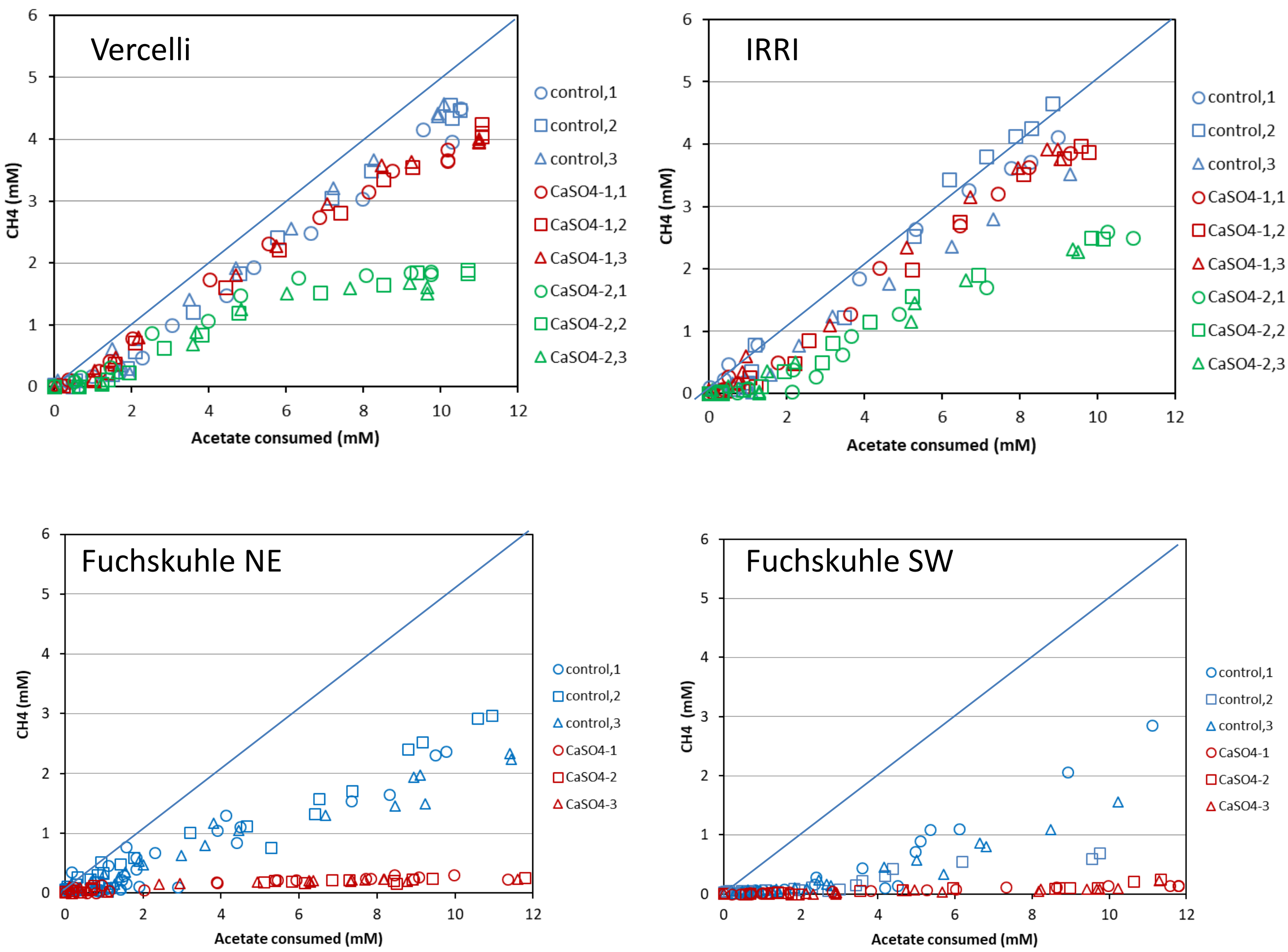

Fig. 5

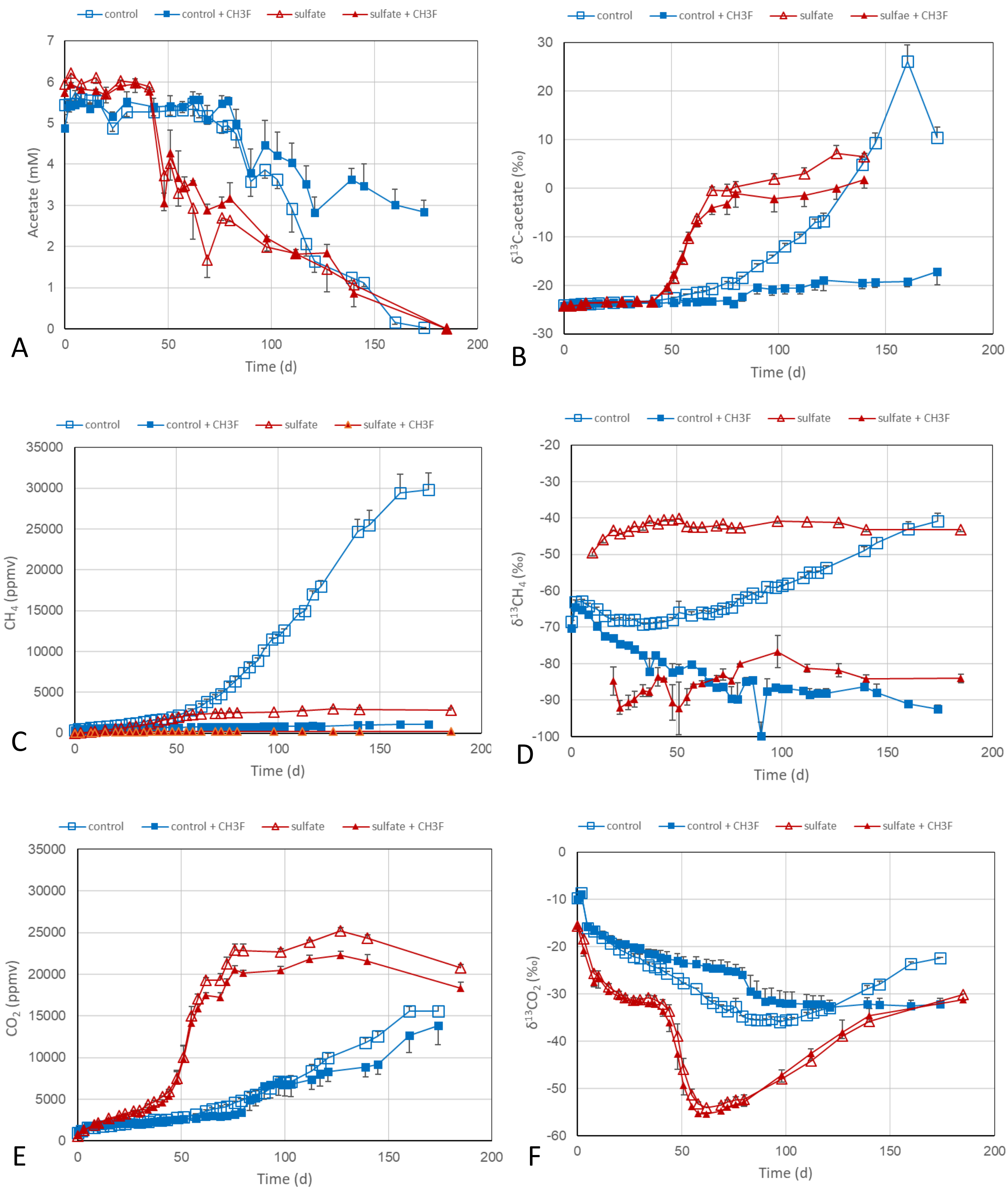

Fig. 6

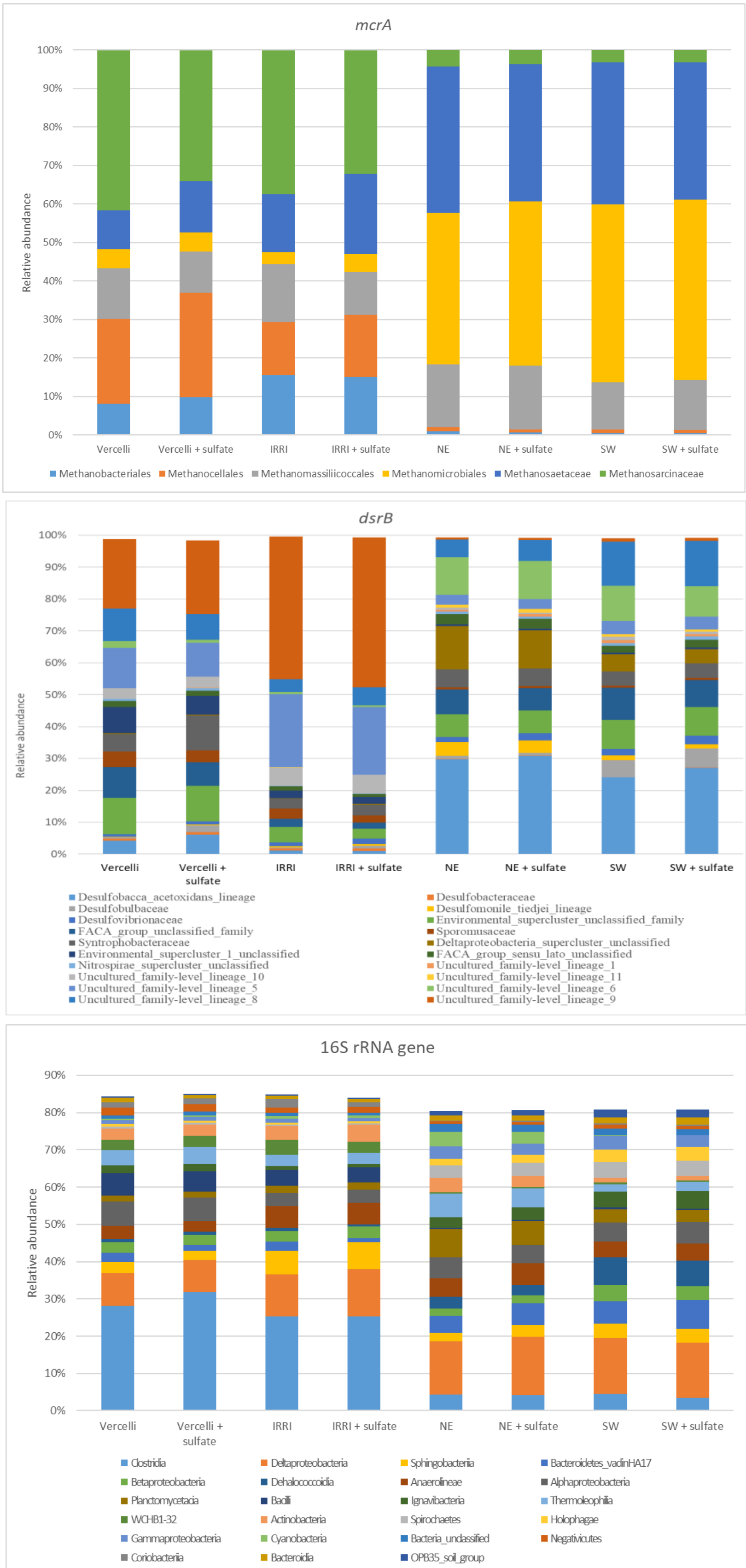

Fig. 7