# Peer review of "Fractionation of stable carbon isotopes during acetate consumption by"

_Biogeosciences, 2021_

## Author Response (AR1)

**Response to the reviewers**

**Reviewer 1**

The submitted manuscript provides a substantial contribution to the field of carbon isotope fractionation of acetate in a number of anoxic environments and the competition for acetate between methanogenic and sulfidogenic microbial communities. The chosen approach is scientific sound and great detail has been used in experimental design and including appropriate control systems. The gained results, e.g. comparing obtained carbon isotope fractionation factors with microbial communities are in great agreement.

As pointed out by the authors, pure culture as well as environmental studies on carbon isotope fractionation during acetate consumption are quite limited, particularly for sulfate-reducing bacteria, and the submitted work provides additional insights in potential processes and pathways involved.

The manuscript is a great fit for the scope of Biogeosciences, covering for example the subject areas biodiversity and ecosystem function, environmental microbiology, biogeochemistry and global elemental cycles, and biogeochemistry and gas exchange.

**Reply:** We thank the reviewer for the positive evaluation.

**Reviewer 2**

The manuscript investigated the acetate consumption under methanogenic or sulfidogenic conditions using carbon isotope fractionation and microbial composition in paddy soils and lake sediments. The main finding is that the magnitude of the isotopic enrichment factors well support the quantification of the methanogenic pathways. However, the microbial compositions could hardly serve for the prediction of the magnitude of enrichment factors. The study system and experimental approach are interesting, the manuscript reads well, the methods are state of the art, and the interpretation of the results are good. It seems little is known about the enrichment factors for aceticlastic methanogenesis and their link between enrichment factors of acetate consumption and metahnogenic, sulfidogenic microbial communities in environmental samples. The lack of understanding justifies by the experiments, thus I like the general concept. The detailed comments are as follows:

**Reply:** We thank the reviewer for the positive evaluation.

Materials and methods:

Line 81: the sampling time should be 2016.

**Reply:** This was a misspelling and will be replaced (20016 by 2016).

Line 119: please revise (iv) 5 ml 50 mM sodium acetate + 4.5 ml $CH_3F$

**Reply:** Again a misspelling, which will be replaced ($CH_3$ by $CH_3F$).

Line 135-146:  Please add references for the primers used here.

**Reply:** The primers were briefly described in the text, with reference to a previous publication, in which primers and references are listed in detail in a table. We think that repeating here all the relevant references for primers would make the presentation too bulky. Therefore, we added a sentence pointing out where information about the primers is found: "The amplification of 16S rRNA, *mcrA* and *dsrB* gene were done as described previously (Liu and Conrad, 2017). This publication also shows in Table S9 all the primers used."

I am a little bit confused about microbial composition analysis. please indicate how many replicates were used for the paddy soil and lake sediments. For the lake sediments, why used the original sediment not the incubated sediment? Mybe it is necessary to explain why used different temperatures for paddy soil and lake sediments incubation.

**Reply:** One replicate was used for each paddy soil and each lake sediment both with and without preincubation with sulfate. We now mention this fact explicitly in the Methods section. Because of only one replication we did not attempt any statistical analyses concerning microbial data. This is mentioned in Results section 3.4. In contrast to paddy soils, which must be incubated under flooded conditions to initiate methanogenesis, incubation is not necessary for the lake sediments, which were methanogenic right from the beginning and thus, could be used directly for the experiments. The in-situ temperatures of paddy soils and lake sediments were different and therefore justified different incubation temperatures (now emphasized in the Methods section).

Results and discussion:

Line 234: the enrichment factors were calculated from the Mariotti plots. The authors should explain why the enrichement factors for NE- and SW-buffer did not show in Fig. 3

**Reply:** In fact, the experimental data are all shown in Fig.3. Note, however, that for lake sediments one set of incubation conditions (the grey and the yellow bars) were not performed.

Line 335: the composition of methanogenic community in the absence and presence of sulfate was quite different between the lake sediment and paddy soil. For instance, in the paddy soil, the relative abundance of Methanosaetaceae and Methanocellales increased, while the relative abundance of Methanosarcinaceae decreased in the presence of sulfate. In particular, methanocellales harbored the same trend with Syntrophobacteraceae in the Vercelli soil. These results should be mentioned in the discussion. Any correlations between this phenomenon and enrichment factors?

**Reply:** We appreciate the reviewer´s comment on the trends in the composition of the methanogenic communities upon sulfate addition. However, we still think that these trends were only marginal (as stated in the manuscript) and in addition, they were not verified by statistical analysis, e.g. correlation analysis (now mentioned). We were interested in effects of the microbial community composition on isotopic enrichment factors. However, such effects were hardly seen, even when comparing flooded soil and lake sediments, which had completely different microbial community compositions but similar fractionation factors.

**Associate Editor:**

I would suggest the authors (i) shorten the abstract and conclusion and probably increase the length of introduction to give more relevant background; and (ii) refine Figs or maybe move some less

important to SI.
What above are optional suggestions, accept or not does not change the status of acceptance of this ms. It contains some useful dataset, thanks.

**Reply:** We shortened the Conclusion, but did not see a possibility to shorten the Abstract without removing important information. We added a sentence to the Introduction to mention previous analysis of methanogenic and sulfidogenic microbial communities in the paddy soils and lake sediments used in the present study. Concerning the Figs, we do not know which kind of refinement we should apply. Since Biogeosciences is an online journal, we think that space should not be a big limiting factor for the presentation of figures. Therefore, we opt keeping all the figures instead of putting some of them into the supplementary information.